# The Potential Role of C-Reactive Protein in Metabolic-Dysfunction-Associated Fatty Liver Disease and Aging

**DOI:** 10.3390/biomedicines11102711

**Published:** 2023-10-05

**Authors:** Zheng Ding, Yuqiu Wei, Jing Peng, Siyu Wang, Guixi Chen, Jiazeng Sun

**Affiliations:** 1Key Laboratory of Precision Nutrition and Food Quality, Department of Nutrition and Health, China Agricultural University, Beijing 100190, China; 2College of Biological Sciences, China Agricultural University, Beijing 100193, China

**Keywords:** C-reactive protein, metabolic-dysfunction-associated fatty liver disease, nonalcoholic fatty liver disease, aging

## Abstract

Nonalcoholic fatty liver disease (NAFLD), recently redefined as metabolic-dysfunction-associated fatty liver disease (MASLD), is liver-metabolism-associated steatohepatitis caused by nonalcoholic factors. NAFLD/MASLD is currently the most prevalent liver disease in the world, affecting one-fourth of the global population, and its prevalence increases with age. Current treatments are limited; one important reason hindering drug development is the insufficient understanding of the onset and pathogenesis of NAFLD/MASLD. C-reactive protein (CRP), a marker of inflammation, has been linked to NAFLD and aging in recent studies. As a conserved acute-phase protein, CRP is widely characterized for its host defense functions, but the link between CRP and NAFLD/MASLD remains unclear. Herein, we discuss the currently available evidence for the involvement of CRP in MASLD to identify areas where further research is needed. We hope this review can provide new insights into the development of aging-associated NAFLD biomarkers and suggest that modulation of CRP signaling is a potential therapeutic target.

## 1. Introduction

Population aging is a worldwide concern and has significant implications for public health, social welfare, and economic development. Aging is a complex biological process that involves the gradual deterioration of various physiological functions and the accumulation of molecular and cellular damage over time [1]. A common feature of senescence at the cellular level is the reduced ability of cells to resume proliferation, which ultimately leads to dysfunction of organs and individuals [2].

As the largest metabolic organ in the body, the liver plays a key role in lipid synthesis and storage. However, liver disease is a leading cause of death, and metabolic-dysfunction-associated steatotic liver disease (MASLD) is one of the most common liver diseases [3]. According to the 2022 World Health Organization report, obesity and overweight are important independent risk factors for MASLD, affecting about 60% of adults and 33% of children in Europe [4]. Despite extensive investigation, there are currently no drugs directly indicated for the treatment of MASLD [5]. The scarcity of reliable biomarkers and noninvasive diagnostics for assessing disease progression is a challenge for MASLD drug development [6]. MASLD is a new term that replaces nonalcoholic fatty liver disease (NAFLD), since factors related to metabolic dysfunction are a more accurate descriptor than the term “nonalcoholic” [7]. Results of a recent study based on past reports of thousands of community subjects showed little difference between MASLD and NAFLD in the general population [8]. A study of 2187 NAFLD patients with imaging-based noninvasive testing proved that at least 98.4% of patients met the definition of MASLD [9]. Therefore, the results of previous NAFLD studies may still be valid under the new definition of MASLD. Although the populations of the two closely matched, there were still individual differences, and both MASLD and NAFLD are retained in this review.

NAFLD shares molecular mechanisms with the aging process, such as oxidative stress, mitochondrial dysfunction, telomere shortening, DNA damage, epigenetic alterations, and cellular senescence. Aging is a major risk factor for NAFLD, evidenced by the rising prevalence of NAFLD with older age [10,11]. Extensive studies investigated the biomarkers related to aging and NAFLD, and some biomarkers with correlations have been detected [12,13,14]. One of the biomarkers that play a potential role in the relationship between aging and NAFLD is C-reactive protein (CRP) [15,16,17,18]. CRP is a protein produced by hepatocytes in response to inflammation and is influenced by aging [19]. CRP may mediate the effects of aging on NAFLD by participating in the senescence process of hepatocytes. Therefore, this review specifically focuses on the direct and indirect regulation of NAFLD/MASLD in aging by CRP and provides a potential target for therapeutic intervention in NAFLD.

## 2. Methods

This review is based on literature searches in the Pubmed database (https://pubmed.ncbi.nlm.nih.gov/ (accessed on 31 August 2023)). The keywords used for the literature search included the following: (high-sensitivity) C-reactive protein, non-alcoholic fatty liver disease, metabolism-related fatty liver disease, aging, cellular senescence, leptin, insulin, mitochondria, reactive oxygen species, signaling pathways, drugs, and targeted therapy. The main summary content included methods, objects, and results that could supplement the relevant content of this review. Our review focuses on the mechanism discussion of CRP and MASLD. In order to gain a deeper understanding of the mechanism of action of these signaling pathways, a search for other associated terms was carried out, for example, receptors of signaling pathways, effector molecules, and transcription factors.

## 3. Aging Is a Risk Factor for NAFLD/MASLD

### 3.1. Pathophysiology of NAFLD/MASLD

NAFLD is a condition where excess fat builds up in the liver, and the diagnosis of NAFLD requires liver steatosis of ≥5% without concurrent heavy alcohol usage [20]. The two forms of NAFLD are nonalcoholic fatty liver (NAFL) and nonalcoholic steatohepatitis (NASH), and NASH can further develop into cirrhosis and hepatic carcinoma. NAFLD is often associated with obesity, insulin resistance (IR), type 2 diabetes (T2D), and metabolic syndrome [21]. NAFLD was initially attempted to be explained by the “two-hit” hypothesis: (1) steatosis leading to susceptibility and fragility of the liver; (2) injury, inflammation, and oxidative-stress-induced exacerbation of steatohepatitis [22]. The “two-hit” hypothesis has now been developed into the “multiple-hit” hypothesis, which includes genetic factors and the influence of the intestinal microenvironment, providing a more accurate explanation for the pathogenesis of NAFLD [23].

The molecular mechanisms of NAFLD are complex and involve multiple factors that affect the metabolism, transport, and storage of lipids in the liver [6]. The essential factors that contribute to NAFLD are (1) increased de novo lipogenesis (DNL) stimulated by glucose and insulin, leading to fatty acid synthesis; (2) impaired beta-oxidation due to mitochondrial dysfunction, oxidative stress, and lipotoxicity, leading to fatty acid accumulation; (3) enhanced lipid uptake from the circulation mediated by transporters and receptors, leading to fatty acid influx; and (4) reduced very-low-density lipoprotein (VLDL) secretion due to endoplasmic reticulum (ER) stress and insulin resistance, leading to fatty acid retention [24]. These pathways are modulated by various factors that affect NAFLD development and progression.

After consensus was reached that NAFLD is not suitable for diagnosis, the nomenclature has been updated to metabolic-associated fatty liver disease (MAFLD) or metabolic-dysfunction-associated steatotic liver disease (MASLD), a term that better describes liver disease associated with metabolic dysfunction [20,25], and the more serious form of MASLD is named metabolic-associated steatohepatitis (MASH) to replace NASH. Patients with MASLD who consume greater amounts of alcohol a week were categorized by a newly created term “MetALD” [25]. The updated nomenclature proved to be more practical for identifying fatty liver patients at high risk of disease progression, as reflected in MAFLD patients with older age, higher BMI levels, and higher rates of metabolic comorbidities [26].

### 3.2. Aging-Associated Impaired Lipid Metabolism and NAFLD/MASLD

Aging is a multifaceted biological phenomenon that results from the progressive decline of various bodily functions and the buildup of molecular and cellular defects over time. Cell senescence is the process of biological aging and is characterized by several hallmarks, such as changes in gene expression and epigenetics, production of reactive oxygen species (ROS), secretion of pro-inflammatory factors, and alterations in metabolism [27]. Liver aging is the result of the accumulation of senescent hepatocytes, where the number of hepatocytes decreases, the regenerative capacity of the liver is impaired, and polyploid hepatocytes accumulate [28]. Aging leads to increased triglyceride levels and a decreased ability to utilize and break down triglyceride (Figure 1) [29], which are important causes of NAFLD. A recent review has discussed impaired liver regeneration in aging mice, and the factors are divided into extracellular (growth factors and cytokines and their receptors) and intracellular (proliferation-associated epigenetic alterations) factors [30]. Furthermore, the senescence-associated secretory phenotype (SASP) accumulates during aging, resulting in decreased autophagy. Senescent cells upregulate pathways such as Bcl-xL, PI3K/AKT, and p53/p21/serpine that prevent or inhibit the activation of apoptosis [30]. The accumulation of senescent hepatocytes and generation of SASP accelerate the development of chronic liver diseases such as NAFLD in the elderly [31].

Recent studies have highlighted the relevance of aging in the progression of NAFLD. A population study showed that the incidence of fatty liver increased significantly from the age of less than 30 to more than 60 years old [32]. Another study pointed out that diabetes, hypertension, and triglycerides ≥150 mg/dL were associated with NAFLD in older adults [33]. It is estimated that 88% of older adults with NAFLD have normal alanine transaminase (ALT) levels, and nearly a quarter of older NAFLD patients are non-obese, which may lead to inaccurate diagnosis of NAFLD. For patients already diagnosed with NAFLD, aged people have a higher tendency for advanced fibrosis than younger people, and fibrosis may contribute to a lower degree of steatosis in patients by limiting fat accumulation [34]. Similar to the results of human studies, 22-month-old mice gained weight accompanied by lipid accumulation and glucose intolerance compared to 3-month-old mice [35]. Another rodent-based study compared high-fat, high-sugar, and high-cholesterol diets to model NAFLD in young and aged mice [36]. Their results showed that aged mice developed NASH, including hepatic steatosis, lobular inflammation, and hepatic ballooning, with a more severe phenotype than younger mice. A study using aged mice and ROS modeling of aged HepG2 cells (a hepatocellular carcinoma cell line) revealed increased lipid accumulation and increased cholesterol and glucose uptake in aged livers [37]. To examine the direct effect of cellular senescence on hepatic steatosis, a study of middle-aged mice and human liver biopsies showed that the accumulation of senescent cells induced hepatic lipid accumulation and steatosis [38]. Their study also revealed that the removal of these senescent hepatocytes reversed the progression of steatohepatitis.

Taken together, current research supports that aging leads to increased incidence and poorer prognosis of NAFLD. Aging-induced changes in the liver include direct effects of reduced lipid metabolism and indirect effects through decreased clearance in senescent cells. Both directly and indirectly, aging and NAFLD are associated with the generation of oxidative stress and inflammation, which is discussed in detail in the next section.

## 4. CRP Is a Risk Factor for Both Aging and NAFLD/MASLD

### 4.1. CRP Definition, Expression Regulation, and Function

In 1930, Tillett and Francis identified a serum protein that formed a precipitate with the C-polysaccharide of pneumococcus, and they called it C-reactive protein [39]. CRP was originally identified as an acute-phase protein because a rapid increase in CRP is found at sites of infection or inflammation.

Hepatocytes are the main parenchymal cells in the liver, making up about 80% of total liver cells [40]. Hepatocytes participate in liver immunity as antigen-presenting cells and activate innate immunity through secreting immune molecules such as CRP [41,42]. The production of CRP by hepatocytes is mainly regulated by cytokines, especially interleukin-6 (IL-6) [43,44]. IL-6 activates the Janus kinase/signal transducers and activators of transcription (JAK/STAT) signaling pathway of hepatocytes, which leads to the transcription of the CRP gene. IL-6 can also induce the expression of CCAAT/enhancer-binding protein β (C/EBPβ), a transcription factor that binds to the CRP promoter and enhances its activity [45]. Interleukin-1β (IL-1β) can enhance the effect of IL-6 on CRP synthesis by enhancing the expression of the IL-6 receptor and C/EBPβ [46]. Interleukin-10 (IL-10) can inhibit the CRP-induced tissue factor gene expression in peripheral blood mononuclear cells [47]. CRP mRNA can be translated into a polypeptide chain of 206 amino acids, which contains a signal peptide of 18 amino acids at the N-terminus [48]. CRP then undergoes processing and folding in the endoplasmic reticulum and moves to the Golgi apparatus, where it is sorted into vesicles and secreted into the blood circulation [49].

CRP is a pentameric protein (pCRP) belonging to the pentraxin family of proteins, which have a characteristic ring-shaped structure composed of five identical monomeric human CRP (mCRP) subunits [50]. Each subunit has a molecular weight of about 23 kDa and contains a phosphocholine (PCh)-binding site that can bind to certain molecules on the surface of dead or dying cells and some types of bacteria [51]. In an inflammatory environment, pCRP undergoes structural changes through association with cell-produced vesicles but still maintains the structural symmetry of the pentamer; the resulting structures, called pCRP*–microvesicle complexes, enhance immune cell recruitment to the site of inflammation [52]. In recent decades, CRP was recognized as an acute-phase reactant that increased in response to various inflammatory stimuli, such as infections, injuries, and autoimmune diseases [53]. CRP binds to phosphocholine moieties on microbial and necrotic cells, initiates the complement cascade, engages Fc receptors to induce pro-inflammatory cytokine secretion, and regulates apoptosis, phagocytosis, nitric oxide synthesis, and coagulation [54,55].

CRP can bind to a specific set of ligands and receptors, such as CD16 (Fc gamma receptor III), CD32 (Fc gamma receptor II), and CD64 (Fc gamma receptor I) [56]. CD64 is one of the main candidates as a therapeutic target for inflammatory macrophage antibodies because it was found to be upregulated in a pro-inflammatory environment [57]. CRP activates macrophages to secrete tissue factor, which plays a procoagulant function [58]. Tissue factor contributes to disseminated intravascular coagulation and thrombus formation under inflammatory conditions. CRP also increases the uptake of low-density lipoprotein (LDL) by macrophages and the ability to form foam cells [59]. In other cases, evidence suggests that CRP may play a protective role to some extent in the excessive activation of inflammation. CRP binding to histone H4 can be a protective response to excessive inflammation and downregulates the neutrophil respiratory burst response [60]. CRP also reduced the excessive activation of the complement system to prevent drug-induced liver damage in mice [61]. Therefore, the function of CRP in immune regulation is complex, and further studies are needed.

### 4.2. The Link between CRP, NAFLD/MASLD, and Aging

Recently, high-sensitivity C-reactive protein (hs-CRP) was proposed as an independent clinical feature of the severity of NASH and the fibrosis caused by NASH, although there are still controversies. A systematic review based on 51 studies evaluated the correlation between 19 different inflammatory cytokines and NAFLD, of which 49 studies included CRP and no more than 12 studies included other cytokines [62]. These studies included more than 36,000 patients, and correlation analyses showed that elevated concentrations of CRP, IL-1β, IL-6, TNF-α, and ICAM-1 were significantly associated with increased risk of NAFLD. The results of this review also show that CRP, but not other inflammatory factors, was simultaneously associated with NAFLD in analyses of serum, plasma, blood, and liver biopsies [62]. Masato’s team examined the levels of hepatic CRP in 100 histologically confirmed NAFLD patients by means of hsCRP and PCR, and the results showed that hsCRP can not only distinguish NASH from pure non-progressive steatosis, but also indicate liver fibrosis in NASH case severity, even after adjusting for age, sex, presence of diabetes, body mass index, and visceral fat [63]. Zhu’s group conducted a study based on the new nomenclature MAFLD, examining the relationship between serum and hs-CRP, and they demonstrated that serum hsCRP levels were positively associated with MAFLD risk, hepatic steatosis, and fibrosis severity in obese Chinese patients [64]. But some studies could not find a link between hsCRP and the degree of hepatic steatosis, necroinflammation, and fibrosis [65]. hsCRP appears to be more associated with obese patients with NAFLD, and hs-CRP levels increase by 19–20% for every 10% increase in BMI [66]. Therefore, the association between CRP and MAFLD prevalence and disease progression may be complex depending on the genetic background and study population. Nonetheless, these studies shed light on the need for a method to measure liver CRP levels in situ and suggest that elevated CRP is associated with fat accumulation leading to NAFLD. Furthermore, the diagnosis of MAFLD/MASLD-defined fatty liver disease based on a spectrum of metabolic syndromes including obesity may make CRP more applicable since both CRP and fatty liver are associated with metabolic disorders.

Coincidentally, the close relationship between CRP and aging was recognized earlier [67,68]. Increased CRP was associated with an increased risk of adverse aging outcomes in a large, nationally representative follow-up study of older adults in England for up to 10 years [18]. A study of 6060 healthy people showed that serum hs-CRP increased significantly with age [69]. A study of 1000 Eastern Europeans aged 65 and over showed an age-dependent increase in IL-6 and CRP levels, and higher IL-6 and CRP levels were associated with poorer physical and cognitive performance [70]. The association remained significant after adjusting for age, sex, BMI, blood lipids, glomerular filtration rate, and smoking status. Identifying CRP levels as an additional parameter in aging assessment improved the performance of the Healthy Aging Index (HAI) in identifying the healthiest older adults in a 10-year follow-up of 934 older adults aged ≥60 years [71]. Common aging-related disease levels including cardiovascular disease, hypertension, and diabetes are associated with elevated CRP, all of which are risk factors for NAFLD [17]. By binding to its receptors, CRP directly or indirectly activates transforming growth factor-beta (TGF-β)/Smad3 and non-TGF-β/Smad3 signaling pathways, induces inflammation and fibrosis, impairs proliferation, and promotes aging [17]. It is worth mentioning that with increasing age and development of NAFLD, an increase in CRP strongly predicted poor prognosis [72,73]. Therefore, it is reasonable to speculate that CRP increases with age to be a potential driver of aging-associated NAFLD.

## 5. CRP Is Involved in the “Multiple-Hit” Mechanisms of NAFLD/MASLD

### 5.1. CRP, Leptin Signaling Pathway, and NAFLD/MASLD

The nomenclature of MASLD emphasizes the central role of metabolic dysfunction in the pathogenesis of fatty liver, in which CRP affects the regulation of fat metabolism by leptin, forming the first hit. Leptin is a 16 kDa pleiotropic peptide hormone encoded by the obesity (Ob) gene that acts by binding to the Ob receptor (ObR) to regulate appetite and energy expenditure [74]. Leptin limits triglyceride storage to prevent lipotoxicity, exerts antisteatogenic effects, and improves insulin sensitivity by inhibiting hepatic glucose production and lipogenesis [75]. Leptin shows a dual role in the development of NAFLD, preventing hepatic steatosis in healthy individuals and early in the disease, but it may act as an inflammatory and fibrotic factor when the disease persists or progresses [76].

Exogenous leptin increased leptin levels without causing weight loss, a phenomenon known as “leptin resistance”, and the hypothesis of an interaction between leptin and plasma circulating factors was later demonstrated [77]. Based on a study of 100 healthy volunteers, after adjustment for factors such as age and BMI, leptin was still independently associated with CRP [78]. Studies based on the interaction between serum proteins and leptin show that human CRP directly inhibits the binding of leptin to the receptor in vitro and blocks the signaling pathway of leptin, which suggests that CRP can promote obesity and metabolic complications [77]. This study further examined whether leptin could stimulate CRP expression in hepatocytes and demonstrated that leptin-induced hepatic CRP production is a PI3K-dependent process. A recent molecular docking study found that in addition to directly binding to leptin molecules, mCRP also docked with the extracellular domain of leptin receptors in a different way of binding to leptin, but the effects on leptin signaling are not yet clear [79]. Disruption of leptin-CRP and the interaction between CRP and leptin receptors may be a target for the treatment of obesity and obesity-related NAFLD.

### 5.2. CRP, Insulin Signaling Pathway, and NAFLD/MASLD

Another pathway involved in lipid metabolism and related to leptin signaling and MASLD is insulin signaling, where CRP forms a second hit in MASLD. Insulin plays an important role in metabolic syndrome during aging, and defects in the regulation of insulin signaling lead to a variety of metabolic diseases, including NAFLD [80,81]. Insulin resistance is a common disorder defined as a state of decreased insulin responsiveness [82]. Individuals with insulin resistance fail to suppress hepatic glucose production and instead increase hepatic lipid synthesis, leading to hyperglycemia and hypertriglyceridemia [83]. Insulin resistance and CRP have been used together as indicators of NAFLD [84]. Elevated CRP is associated with insulin resistance [85]. CRP has been shown to inhibit insulin signaling through Fcγ, and genetic elimination of CRP confers resistance to obesity and insulin resistance in rats [86,87]. Human recombinant CRP may increase IRS-1 phosphorylation through JNK and ERK1/2, leading to insulin resistance and increased glucose uptake [88]. Insulin-like growth factor 1 (IGF-1) was shown to be inversely correlated with CRP in previous studies [89,90,91]. IGF-1 mRNA levels in the liver and brain decline with age [92]. Low serum IGF-1 levels were associated with increased histologic severity of NAFLD in a cross-sectional study [93]. Reductions in IGF-1 and IGF-binding protein were associated with increased NAFLD severity [94]. Insulin initiates phosphorylation events that activate phosphoinositide 3-kinase (PI3K), a lipid kinase that coordinates glucose uptake and utilization [95]. IGF-1 was proved to block CRP by activating the PI3K/Akt pathway and inhibiting the c-Jun N-terminal kinase (JNK) signaling pathways [96]. Therefore, an antagonistic relationship between CRP and IGF-1 needs further verification.

AMP-activated protein kinase (AMPK) signaling suppresses insulin resistance, promotes glucose metabolism and uptake, and has been shown to alleviate diabetes [97]. In NAFLD treatment, AMPK inhibits fatty acid synthesis by downregulating the expression of fatty acid production genes and increases the expression of genes involved in fatty acid oxidation [98]. Studies in obese mice also proved that hepatic AMPK activation is effective in preventing steatosis and inflammation [99]. However, high levels of CRP inhibit AMPK signaling [100], and AMPK activation decreases with age, causing impaired metabolism, increased oxidative stress, and reduced autophagic clearance [101]. Therefore, in addition to directly affecting the insulin signaling pathway, CRP can also indirectly reduce the function of insulin by inhibiting AMPK signaling.

The hypothesis that CRP affects NAFLD by affecting the insulin pathway was further substantiated in the mammalian target of rapamycin (mTOR), one of the downstream effectors of IGF-1 and AMPK. mTOR regulates cell growth and metabolism in response to nutrients, growth factors, and cellular energy [102]. mTOR has been identified as a central node in a network regulating hepatic lipid metabolism, but its specific impact on NAFLD is unclear [103]. A mouse model of spontaneous HCC reveals that mTOR crosstalk with STAT5 promotes de novo lipid synthesis and induces HCC [104]. A study using CRP transgenic mice has shown that CRP activates Smad3/mTOR signaling through TGF-β and ERK/mitogen-activated protein kinase (MAPK) [105]. Collectively, CRP shows inhibitory effects on the insulin signaling pathway, and increased CRP with age is a potential biomarker and cause of insulin resistance and NAFLD.

### 5.3. CRP, Mitochondrial Dysfunction, and NAFLD/MASLD

The third hit of CRP in MASLD is inducing ROS production and leading to the impairment of mitochondrial function. Mitochondrial dysfunction is a condition in which mitochondria fail to perform their functions of cellular respiration and signaling. Insulin resistance underlies the development of type 2 diabetes mellitus (T2DM), where mitochondrial dysfunction is one of the main mechanisms [106]. Patients with T2DM are prone to develop NAFLD and NASH [107]. ROS are byproducts of mitochondrial aerobic respiration, causing oxidative damage to DNA, proteins, and lipids [108]. Cellular senescence leads to decreased mitochondrial ATP synthesis and ROS accumulation, which leads to an increase in ROS with age [109]. ROS damage mitochondrial membranes, leading to MPTP formation and subsequent release of mtDNA as danger-associated molecular patterns (DAMPs), which activate the NLRP3 inflammasome to trigger inflammation through toll-like receptor (TLR) signaling [110]. CRP upregulates ROS production in target cells through Fcγ receptors 114. CRP enhances ROS formation and innate killer mechanism phagocytosis in a complement-dependent manner in patients with sepsis [111]. Radiation-induced CRP was proven to interact with the STAT3/Ref-1 complex through ROS, resulting in further mitochondrial dysfunction [112]. ROS overproduction suppresses the ability of antioxidant defense systems in NAFLD, leading to further oxidative damage [113]. Low mitochondrial DNA (mtDNA), important for cellular homeostasis, was associated with higher CRP levels in healthy adults [114]. Another population-based study also showed that participants with higher levels of CRP and IL-6 had lower blood mtDNA levels [115]. ROS-induced mitochondrial damage is an important cause of mitochondrial DNA release, exacerbating mitochondrial dysfunction and chronic inflammation [116].

### 5.4. CRP, NF-κB Pathway, and NAFLD/MALSD

The production of inflammatory factors through activation of the NF-κB signaling pathway is the fourth hit of CRP in MASLD. NF-κB is widely recognized as an important regulator in aging and inflammation, and it is essential for many important immune transcriptional programs, including the response of innate immune cells to pathogenic microorganisms. NF-κB acts as a transcription factor that regulates multiple aspects of immune function and plays a central role in inflammatory responses [117]. NF-κB transcription increases with age and is associated with age-related degenerative diseases, and pathways such as insulin/IGF-1 FoxO and mTOR are correlated with NF-κB signaling [118]. Activation of NF-κB promotes the expression of inflammatory cytokines, such as TNF-α, IL-1, and IL-6, which in turn enhance NF-κB activity [119]. NF-κB activation is shown to be increased in the liver of NAFLD patients and has a central role in the regulation of hepatitis fibrosis and carcinogenesis [120]. The results of monocytes incubated with CRP showed that CRP treatment led to an increase in the number of M1 inflammatory macrophages and mediated the production of inflammatory cytokines, while CD32/CD64 small interfering RNA or dominant negative NF-κB blocked the effects of CRP [121]. CRP activates the NF-κB signaling pathway in endothelial cells through the degradation of IκB-α [122]. A study of CRP-driven diabetic nephropathy showed that CRP triggers the aggregation of DPP4 and CD32b at the protein level to form DPP4/CD32b/NF-κB signaling [123]. Studies in ARPE-19 cells revealed that CRP mediates IL-8 production and triggers inflammation via Fcγ/NF-kB and MAPK pathways [124]. NF-kB is critical for intracellular signaling induced by CRP pro-inflammatory and catabolic mediators [125]. NF-κB activation and CRP expression are region-specific in response to endotoxemia, and the NF-κB transcription factor subunit p50 interacts with consensus sequence elements of CRP promoter, which implies a cyclic relationship between CRP and NF-κB [126].

Although the mechanisms are not fully understood, leptin signaling, insulin signaling, mitochondrial metabolism, and the NF-κB pathway, all involving CRP, are the potential factors that modulate NAFLD/MASLD (Figure 2). These pathways not only are signaling pathways involving CRP, but also are known to be important in the occurrence and development of NAFLD/MASLD. On the one hand, CRP mediates the impairment of lipid metabolism by inhibiting the leptin and insulin signaling pathways. On the other hand, CRP takes advantage of the interaction between liver cells and Kupffer cells to increase levels of itself, and the CRP-mediated pro-inflammatory factors further promote the development of inflammation, forming the “multiple hits” involved in NAFLD/MASLD mechanisms.

## 6. Potential Strategies to Lower CRP in NAFLD/MASLD Treatment

### 6.1. Drugs Used in NAFLD/MASLD Treatment and CRP Lowering

Some potential drugs used in NAFLD treatment have been shown to reduce CRP. Pioglitazone is a PPARγ agonist that alleviates NAFLD by reducing insulin resistance [127]. Early studies have demonstrated the efficacy of pioglitazone in reducing CRP, which may be mediated through PPARγ-NF-κB signaling [128,129]. As a class of HMG CoA reductase inhibitors, statins are widely used lipid-lowering drugs [130]. Statins reduce the production of IL-6 and reduce the number of LDL particles, thereby reducing the production of inflammatory mediators produced by CRP [131]. Limited cross-sectional studies show that statins are effective in reducing steatosis, inflammation, and fibrosis in NAFLD or NASH [130]. Vitamin E, either in the alpha-tocopherol or gamma-tocopherol form, has been shown to lower serum CRP levels [132]. The mechanism of vitamin E in anti-inflammation and reducing CRP production is also mediated through PPARγ [133]. Vitamin E improves biochemical parameters in NASH patients with histological abnormalities by inhibiting inflammation and reducing ROS [134]. Furthermore, mitochondrial quinone (MitoQ), a mitochondria-targeted antioxidant, can reduce ROS production through the action of NF-κB and lower the risk of cardiovascular disease in type 2 diabetes [135]. Astaxanthin and β-Cryptoxanthin, two strong antioxidants belonging to the group of carotenoids, have been proven to treat NAFLD and aging by resisting oxidative stress, inhibiting inflammation, and promoting M2 macrophage polarization [136]. A meta-analysis showed that higher astaxanthin intake is associated with a decrease in CRP, and no significant associations were observed for other indicators [137]. A study of the link between vitamins and CRP found that only carotene is inversely associated with increased CRP levels [138].

### 6.2. CRP- and CRP-Receptor-Targeted Therapy

Some evidence reflects the possible role of targeting CRP in NAFLD. A study examining the effects of CRP gene ablation on obesity in rats showed that CRP deficiency resulted in significant reductions in body weight and food intake, increased energy expenditure, and improved insulin sensitivity [90]. The study further showed that systemic knockout of CRP promotes leptin glucose metabolism in the liver and skeletal muscle and enhances leptin-stimulated STAT3/Akt signaling. However, the study did not examine whether CRP knockout affects NAFLD-related liver phenotypic changes. Various therapeutic agents are being investigated that may alter levels of inflammatory markers, including CRP. 1,6-bis(phosphocholine)-hexane, a specific molecule inhibitor of CRP, was developed to treat CRP-induced cardiovascular disease [139]. This CRP inhibitor was shown to reduce CRP and attenuate increased infarct size and cardiac dysfunction in a rat model of myocardial infarction [140]. Subsequently, an antisense oligonucleotide (ASO) drug that specifically targets CRP was developed and demonstrated a >70% reduction in CRP and improved carotid artery patency in mice [141], and an ASO drug was shown to dose-dependently reduce CRP in a Phase II study [142]. Selective elimination of dysregulated macrophages to alleviate excessive inflammatory cytokine production has become one of the focuses of next-generation therapies [143]. Targeted drugs for CD64, the main receptor of CRP, have been developed for the treatment of macrophage-mediated chronic inflammatory diseases, such as rheumatoid arthritis and diabetes [58]. Novel CD64-targeted drugs have been shown to specifically eliminate M1 macrophages [144]. Drugs targeting CRP and its receptors are currently under development, and no studies have been conducted for the treatment of NAFLD.

### 6.3. CRP Adsorption Technology

A variety of approaches were tested to target CRP lowering, including the inhibition of CRP synthesis, CRP receptor antagonists, and complement activation inhibitors [145]. However, these methods are limited by low efficiency and side effects caused by non-specific binding. Recently, an exciting technology, namely CRP apheresis, has been published and put into use, [146]. This technology uses a novel adsorbent (PentraSorb CRP) for CRP apheresis in human plasma, specifically binding CRP in human plasma, and can regenerate up to 200 times without losing its binding ability or affecting biocompatibility [146]. Although there are no examples of CRP apheresis applied for the treatment of NAFLD/MASLD, it has been used in the treatment of other acute diseases. A clinical study based on myocardial infarction patients showed that CRP concentration can be effectively reduced by CRP apheresis, and no side effects were observed during the 12-month operation [147]. In addition, CRP apheresis used in acute infections including sepsis and coronavirus infection was shown to be effective in reducing CRP in plasma [148,149]. After further confirmation of safety and necessity evaluation, CRP apheresis is worthy of becoming one of the potential methods for the treatment of NAFLD and other chronic diseases.

## 7. Conclusions

In summary, as an indicator of inflammation that increases with age, CRP can contribute to the development of NAFLD in aging by mediating insulin signaling, mitochondrial metabolism, and NF-κB signaling. In growth and development, metabolism, and apoptosis, CRP is an important bridge connecting chronic inflammation, aging, and NAFLD. Treatment of NAFLD by lowering CRP is promising, but current studies and strategies are limited. This review is the first to propose CRP as a potential regulator for NAFLD in aging. Considering that the nomenclature of NAFLD has changed to MASLD, further investigation is needed to verify whether previous correlations are suitable for the new nomenclature. In addition, future research is needed to explore the deeper mechanism of CRP in MASLD and develop new drugs or treatments based on signaling pathways involving CRP.

## Figures and Tables

**Figure 1 biomedicines-11-02711-f001:**
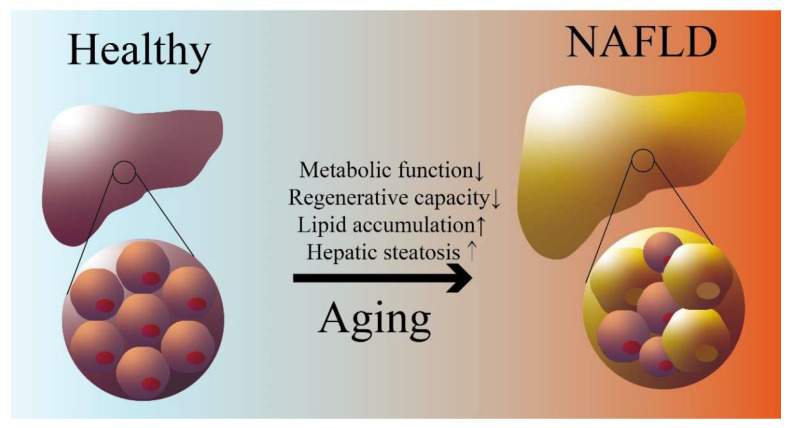
Aging and NAFLD. Impairment of lipid metabolism and liver regenerative capacity with age leads to hepatic lipid accumulation and steatosis. (→ refer to the process of aging).

**Figure 2 biomedicines-11-02711-f002:**
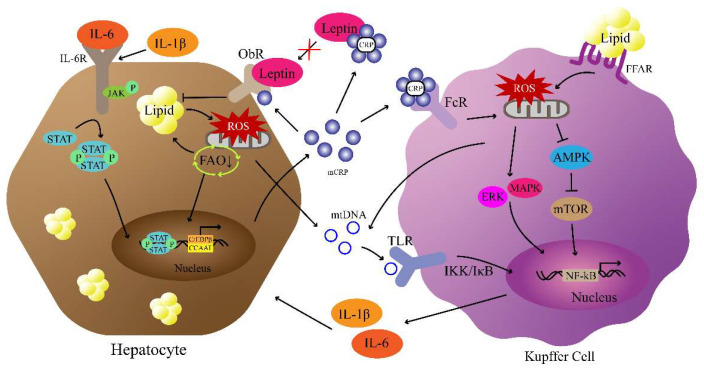
Potential pathways of CRP-mediated NAFLD/MASLD pathogenesis. Increased CRP neutralizes leptin, resulting in lipid accumulation. Lipid accumulation in hepatocytes induces ROS production and mtDNA release. Then, the activation of TLR by mtDNA results in the nuclear transfer of NF-κB to produce IL-6 and IL-1β. Subsequently, the IL-6/JAK/STAT pathway mediates the CRP production of hepatocytes that activates CRP/ROS/NF-κB signaling. (↓ refers to promotion and ┴ refers to inhibition).

## Data Availability

No new data were created or analyzed in this study. Data sharing is not applicable to this article.

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
