# Peer review of "The Potential Role of C-Reactive Protein in Metabolic-Dysfunction-Associated Fatty Liver Disease and Aging"

_biomedicines, 2023, doi:10.3390/biomedicines11102711_

Round 1
Reviewer 1 Report (Previous Reviewer 1)
I thank the Authors for addressing my previous concerns. I'd suggest to revise the final statement regarding the validity of the reported observation for MAFLD. I do not think that the nomenclature change may affect the data reported.
A few sentences would still benefit of a revision, however language has been improved
Author Response
Thank you very much for your suggestion. After consideration, we have added MASLD to each chapter title of the article. MASLD is used because of the recent consensus among liver disease experts. Moreover, we supplement the article with the latest research conclusions on the relationship between NAFLD and MASLD applicability. Several recent studies have examined the population relationship and applicability of MASLD and NAFLD and found that older NAFLD findings are still valid under the new MASLD definition. Although the populations of the two closely matched, there were still individual differences, so we retained both MASLD and NAFLD.
Reviewer 2 Report (Previous Reviewer 2)
The paper has been successfully revised and improved by omitting unnecessary material and adding important subchapters and information.
The paper still needs to present very clearly, that CRP is - up to current knowledge - just one inflammatory player among others and does not show any unique properties over others. It is an unspecific marker for lots of diseases and disease mechanisms. Its predictive potential for CVD, T2DM, NAFLD and other disorders is outranked by a whole battery of other parameters, even though it shows "some" association.
Thus, this paper needs to emphasize, why CRP (and not other parameters) should be focussed on. The chapter on CRP-targetting therapies is crucial by showing which effects can be expected when specifically adressing CRP and taking it out of the equation. Everything else in this review is merely a presentation of CRP as a minute part in a whole network of signals, cells and processes. You need to clarify the outstanding properties of CRP, not the mainstream facts.
moderate changes needed
Author Response
Thank you for your valuable comments and suggestions. In this revision, we highlight the unique mechanisms of CRP and combine these mechanisms with the pathogenesis of MASLD, proposing the “multiple hits” of CRP on MASLD. First, CRP inhibits leptin's regulation of fat metabolism by directly binding to leptin, leading to fat accumulation and metabolic disorders in the liver; second, CRP causes insulin resistance by interfering with the insulin signaling pathway and promotes the occurrence of MASLD; third, leptin and excessive accumulation of lipids caused by impaired insulin signaling, together with CRP-mediated ROS production, lead to mitochondrial damage; fourth, CRP uses the interaction between hepatocytes and Kupffer cells to promote the increase of its own levels, and through NF-κB signaling to produce more inflammatory factors and promote the development of fatty liver. Despite our efforts to increase the content of CRP treatment, but there are few data available for reference, and further research is still needed.
Reviewer 3 Report (Previous Reviewer 3)
Authors submitted a revised version of their manuscript. Authors have addressed most of my comments, however the main point has not been sufficiently supported by published evidence.
Main point of the paper is that authors propose to use CRP as a biomarker of NAFLD in aging population. However, no evidence has been published or mentioned in the review, that support this statement. Authors do not provide a search methodology, so it is unclear if they have reviewed all the available evidence.
First, as authors mention correctly, CRP is increasing with age. Second, it is also proven that patients with clinically proven NAFLD have increased levels of CRP. However this increase may be more probably due to other associated aspects of metabolic syndrome - high cholesterol, atherosclerosis etc) Rather conflicting evidence has been published on the association of CRP with histological NASH (see review by https://www.frontiersin.org/articles/10.3389/fimmu.2020.634409/full).
Also the increase of CRP in aging non NAFLD population creates further complications in assessing steatosis in elderly.
Also authors show some potential mechanisms of association of CRP with NAFLD (e.g. section 3.2 - insulin resistance which is more associated with T2DM, or ROS as a result of hyperglycemia)
If the authors want to provide a meaningful contribution to the field I recommend performing a systematic review and quantitative meta-analysis with the following PECO
P: adult population
E: CRP levels
C: none
O: presence of NAFLD and NASH
Otherwise I could recomend publication with major revision - please do not invent statements in the referenced studies (ref 65) that are not mentioned in said studies. In the actual study (https://www.ncbi.nlm.nih.gov/pmc/articles/PMC6128988/) increase of CRP was associated with hypercholesterolemia and NAFLD was not even examined and there is no mention of NAFLD or liver steatosis in the whole manuscript.
Author Response
Thank you for your valuable comments and suggestions. We mainly searched for articles in the last 20 years and some earlier articles, but the relationship between CRP and NAFLD has been stuck in the correlation research and lacks deeper mechanism research. The method of literature search is to use the pubmed database and enter keywords, such as CRP and NAFLD. The search results show that the document range is from 2004 to 2023. Other search terms include: aging, cellular senescence, leptin, insulin, diabetes, mitochondria, oxidative stress, etc. Most of these related studies are in the last 20 years. We did not introduce all the literature in the article, because the relevant research is massive, but the in-depth mechanism research is only a handful. Our citation of the correlation between CRP and NAFLD is not without consideration of the contradictions and maladaptations that exist, so we supplement these contents in the article.
This article focus on providing references for the mechanism of CRP participating in NAFLD, Because of these factors and the existence of contradictions and the differences in race and living habits between countries and regions, there are still emerging studies pointing out the link between hsCRP and NAFLD (https://www.frontiersin.org/articles/10.3389/fimmu.2022.880298/full), which further shows that the mechanism investigation is needed to explain how CRP participate in NAFLD.
After consideration, we revised the statement of age-related NALFD, because there is indeed no direct evidence that CRP is associated with age-related NAFLD. We very much agree with you that CRP levels are affected by other diseases such as obesity, insulin resistance and cardiovascular disease, and there is currently conflicting evidence. These factors are closely related to the occurrence and development of NAFLD, so that the definition of NAFLD has been changed to MAFLD. Even after the definition was changed, the populations of NAFLD and MASLD remained consistent, which shows that these factors have been included in MASLD (https://www.journal-of-hepatology.eu/article/S0168-8278(23)05000-6/fulltext).
We also want to engage in related research in the future, but before the real mechanism is researched, the existing evidence can only provide ideas for future research and narrow the scope of research. Therefore, we would prefer to conduct relevant studies and write systematic reviews based on meta-analysis in the future.
Thank you very much for pointing out “statements in the referenced studies (ref 65)”. The original intention was just to express that CRP is related to the growth of age, so we revised this sentence.
Reviewer 4 Report (Previous Reviewer 4)
After reviewing the article submitted for review, I consider that it is basically a future line of work, not presenting results that modify the current knowledge of the pathology, therefore I would not accept the article for publication in the journal.
The article would need to be proofread by a native English speaker.
Author Response
Thank you for pointing out that our review does “not presenting results that modify the current knowledge”. In this revision, we combine the effects of CRP on NAFLD and the pathogenesis of NAFLD to propose a "multiple hit" mechanism of CRP. In terms of content, we more objectively describe the conclusions of existing studies and the current contradictions. Our review places greater emphasis on CRP as a player in NAFLD rather than just a biomarker and suggests CRP as a possible therapeutic target.
Round 2
Reviewer 3 Report (Previous Reviewer 3)
thank you for the revisions
Author Response
Thank you very much again for taking time to review this manuscript.
Reviewer 4 Report (Previous Reviewer 4)
After evaluation of the article submitted for review, my assessment remains unchanged. The article evaluates a possible line of work but does not add anything to current knowledge on the subject. Therefore, when the authors have data on the current line of work, I believe that its publication in the journal could be re-evaluated.
No changes in English Language must be done.
Author Response
We appreciate the thoughtful comments of Reviewer 4 . We agree that it “The article does not add anything to current knowledge on the subject” and that “a more rigorous data analysis” is needed. First, a recent systematic review for articles published from 1 January 1960 to 31 December 2021 using electronic databases (https://www.ncbi.nlm.nih.gov/pmc/articles/PMC9122097/) on inflammatory factors and NAFLD has fully demonstrated the correlation between CRP and NAFLD. To emphasize this point, we supplement the content in this review in section 4.2 of the article:
“A systematic review based on 51 studies evaluated the correlation between 19 different inflammatory cytokines and NAFLD, of which 49 studies included CRP and no more than 12 studies on other cytokines. These studies included more than 36,000 patients, and correlation analyzes showed that elevated concentrations of CRP, IL-1β, IL-6, TNF-α, and ICAM-1 were significantly associated with increased risk of NAFLD. The results of this review also show that, CRP, but not other inflammatory factors, was simultaneously associated with NAFLD in analyzes of serum, plasma, blood and liver biopsies”.
Second, the causal relationship between CRP and NAFLD in human has not yet been elucidated, but recent studies on CRP gene elimination in rats alleviated liver fat accumulation have proven that CRP is an important cause of NAFLD. This content is related to the mechanism of CRP-mediated NAFLD and has been placed in section 5.1.
Finally, and the most difficult part, we have tried to obtain data on the age of patients in relevant studies, but perhaps due to issues involving patient privacy and property rights, few study has provided the age, CRP, and degree of NAFLD for each patient.
Therefore, this article is not to further illustrate the causal relationship between CRP and NAFLD in population studies, since inflammation and NAFLD are considered reciprocal. Even if the article does not have a new perspective, we have provided new insights into the pathogenesis of CRP in NAFLD, which is reflected in the role of CRP in the "multiple hits" of NAFLD, which was not mentioned in previous articles.
This manuscript is a resubmission of an earlier submission. The following is a list of the peer review reports and author responses from that submission.
Round 1
Reviewer 1 Report
The Authors of this review address the role of CRP as a biomarker, mediator, and potential treatment target in NAFLD. The issue is not particularly novel; however, the review achieves its intent to supply a comprehensive view of the problem and it may be of help for scientists and clinicians dealing with NAFLD. The reference list is extensive and updated. Table 1 could be omitted or revised as in its present form does not usefully summarize the information of paragraph 4. Figure 2 legend should be shortened and revised to improve its clarity, avoiding reference to data not shown in the figure itself.
Minor points
11. Please carefully revise the use of acronyms. They should always be reported in full at first use (also in figure legends) and use consistently at following mentions. Instead NASH is not explained at line 34, NF-KB is written in full at line 239, but not at line 160 (first use), ROS appears again in full at line 220, when the acronym has already been explained at line 87 etc.
12. The sentence at lines 267-268 needs to be explained.. Compared to which diet does medDiet suggest a lower carbohydrate intake? Med Diet is usually considered a diet with a high carbohydrate intake.
There are several points that need to be revised:
11. Please carefully revise the use of acronyms. They should always be reported in full at first use (also in figure legends) and use consistently at following mentions. Instead NASH is not explained at line 34, NF-KB is written in full at line 239, but not at line 160 (first use), ROS appears again in full at line 220, when the acronym has already been explained at line 87 etc.
22. Line 29: The sentence is incomplete; it should be specified that these changes are a consequence of aging
33. Line 63 the sentence should be revised to avoid repeating “liver” thrice in two lines
44. Line 67 The verb “attribute” seems misused in this sentence, please clarify
55. Line 74 Please add “years old” after 60
66. Line 76 The adverb “soberingly” is not appropriate here, please revise
77. Line 152 please add “study” after “follow-up”
88. Line 154: Which is the subject in the sentence “…was one of the risk factors”? Please revise
99. Lines 157-8 It should read “ Since CRP is produced mainly by hepatocytes…” Also the consequentiality between this and the following sentence is unclear.
110. Lines 158-159 Did you mean ..” that since CRP increases with age, it may represent an important biomarker and driver of aging-associated NAFLD”?
111. Line 182-184 “has been shown”
112. Line 187 The term “diagnosis-based” is unclear, please revise
113. Line 204 Please add “effector” or “mediator” after downstream
114. Lines 276 “higher intake”
115. Line 320 please revise and clarify this sentence. The use of the term “faithful” is unclear and which characteristic of CRP is consistent in different conditions needs to be specified
Reviewer 2 Report
The authors provide a narrative review on the impact of CRP on NAFLD and potential treatment options utilizing CRP action.
The review broadly summarizes aspects of pathophysiology affecting CRP, NAFLD and their association. The article also mentions various treatments, which reduce CRP in parallel to dozens of other parameters.
CRP is neither specific for NAFLD, nor has any specific treatment targetting CRP shown a useful effect. The review therefore does not provide relevant new information.
moderate changes needed
Reviewer 3 Report
The manuscript is a narrative review focused on CRP and its relationship with NAFLD.
The manuscript is inconsistent, some parts are well, written, others rather poorly (e.g. introduction)
1) Overall the manuscript uses outdated terminology, NAFLD is being phased out, replaced with steatotic liver disease that includes metabolic associated steatotic liver disease and others...
2) The definitions are used incorrectly, older terminology meant that NAFLD had two categories - simple steatosis (NAFL) and steatohepatitis (NASH), This is missing completely in the manuscript. NASH is not defined anywhere
3) English - although I am not qualified, in some parts english is so bad that it prevents understanding - e.g. line 64.
4) Abstract - line 19 - this is not a study but a narrative review.
5) Introduction - tries to summarize pathophysiology of NAFLD.
-Insuline resistance is missing completely.
- the fact that CRP is synthesized in the liver does not suggest its direct involvement in NAFLD, because its secreted into the bloodstream and distributed globally.
- authors mention a subcategory of "aging associated NAFLD" that is a novelty to me, please define (e.g. age threshold) and describe differences between "aging associated NAFLD" and "common NAFLD"
6) Aging
- the term senescent hepatocyte is misunderstood, It is not a hepatocyte in older person, but a hepatocyte with specific phenotype that may appear also in young people with liver disease, please refine the section about liver regeneration with better description of the data - e.g. https://www.ncbi.nlm.nih.gov/pmc/articles/PMC6128415/
7) CRP - line 146 - crp should be viewed dialectically - please describe to what this relates? e.g. as pro and also inflammatory cytokine or other?
8) line 217 - "factories of cellular energy synthesis". Energy cannot by synthesized, rewrite correctly
9) line 262 - "strictly restricted diet" restricted in what, elaborate, there are significantly more data on caloric restriction and length of life than that one particular study in mice. Correlations are also in humans.
10) chapter "CRP lowering drugs" is completely nonsense, since no drug is specifically lowering CRP, the drugs described influence inflammation and thus decrease CRP levels as a marker of inflammation. Rephrase/rewrite
Reviewer 4 Report
After reviewing the article submitted for review, I consider that it is basically a review of the possible implications of C-reactive protein on non-alcoholic fatty liver disease. The authors review possible pathogenic pathways in which CRP may play a role, but in my opinion it is a personal opinion to be evaluated in the future for more in-depth studies. Unfortunately, I do not think it would be of interest to publish this paper in the journal in its current format.
English quality is fine.